# Spatial Memory in a Spiking Neural Network with Robot Embodiment

**DOI:** 10.3390/s21082678

**Published:** 2021-04-10

**Authors:** Sergey A. Lobov, Alexey I. Zharinov, Valeri A. Makarov, Victor B. Kazantsev

**Affiliations:** 1Neurotechnology Department, Lobachevsky State University of Nizhny Novgorod, 23 Gagarin Ave., 603950 Nizhny Novgorod, Russia; zharinov.lexa.az@gmail.com (A.I.Z.); vmakarov@ucm.es (V.A.M.); kazantsev@neuro.nnov.ru (V.B.K.); 2Neuroscience and Cognitive Technology Laboratory, Center for Technologies in Robotics and Mechatronics Components, Innopolis University, 1 Universitetskaya Str., 420500 Innopolis, Russia; 3Center For Neurotechnology and Machine Learning, Immanuel Kant Baltic Federal University, 14 Nevsky Str., 236016 Kaliningrad, Russia; 4Instituto de Matemática Interdisciplinar, Facultad de Ciencias Matemáticas, Universidad Complutense de Madrid, 28040 Madrid, Spain; 5Lab of Neurocybernetics, Russian State Scientific Center for Robotics and Technical Cybernetics, 21 Tikhoretsky Ave., St., 194064 Petersburg, Russia

**Keywords:** spiking neural networks, STDP, learning, neurorobotics, cognitive maps, vector field of synaptic connections, vector field of functional connections

## Abstract

Cognitive maps and spatial memory are fundamental paradigms of brain functioning. Here, we present a spiking neural network (SNN) capable of generating an internal representation of the external environment and implementing spatial memory. The SNN initially has a non-specific architecture, which is then shaped by Hebbian-type synaptic plasticity. The network receives stimuli at specific loci, while the memory retrieval operates as a functional SNN response in the form of population bursts. The SNN function is explored through its embodiment in a robot moving in an arena with safe and dangerous zones. We propose a measure of the global network memory using the synaptic vector field approach to validate results and calculate information characteristics, including learning curves. We show that after training, the SNN can effectively control the robot’s cognitive behavior, allowing it to avoid dangerous regions in the arena. However, the learning is not perfect. The robot eventually visits dangerous areas. Such behavior, also observed in animals, enables relearning in time-evolving environments. If a dangerous zone moves into another place, the SNN remaps positive and negative areas, allowing escaping the catastrophic interference phenomenon known for some AI architectures. Thus, the robot adapts to changing world.

## 1. Introduction

All natural creatures at different organizational levels possess a memory, which can be broadly defined as storing and recalling past information states of the organism. Current neurobiological studies of memory focus on the problems of its formation, storage, and representation [1,2,3]. In animals and humans, different types of memories are inextricably linked with learning. As a rule, learning is an iterative process, which yields an increment of skills at each training iteration. The power law of practice describes such training in the form of learning curves [4,5].

In contrast, artificial systems usually rely on one-shot memory and use no subsequent adjustments. The process of storing information in technical systems also differs from neurobiological memory. The memory in humans and animals, in general, is not entirely reliable and wanes with time. To account for this feature, at the end of the 19th century, Ebbinghaus [6] introduced the so-called forgetting curves. The forgetting process occurs most intensively immediately after the information has been received. It turns out that learning curves have a similar dependency pattern as forgetting curves, which may imply inner interaction and interdependence of these processes.

It is believed that the primary biological mechanisms of learning and memory are determined by neuronal plasticity, which is relatively well studied at the level of individual cells and synapses [7]. However, its extrapolation into the neural network behavior is a challenge. It has still been poorly understood because of the high complexity of monitoring brain circuits during functional tasks. The critical point is the processes of structural and functional rearrangements occurring at the global network level, involving local mechanisms of Hebbian-type synaptic plasticity. In particular, such rearrangements have been found in experimental models of cortical and hippocampal cultures in vitro [8,9,10,11] and the CA3-CA1 pathway in vivo after applying a long-term potentiation protocol [12].

Culture neural networks can exhibit simple forms of memory, and this memory is susceptible to elementary learning. In particular, local electrical stimulation of a neuronal culture can lead to changes in its network activity. For example, one can train the network to elicit spikes at the desired electrode in the chosen time window or increase the average response activity [9,10,13]. Note that, similar to learning protocols for animal training, in culture networks, consequent trials lead to decreased stimulation time required to achieve the desired state.

The use of biologically inspired models of spiking neural networks (SNNs) with Spike-Timing Dependent Plasticity (STDP) is a convenient tool for studying the mechanisms of plastic changes, the formation of network structures, and population bursts [14,15,16]. STDP represents Hebbian-type synaptic plasticity, widely experimentally verified [17,18]. According to this mechanism, the synaptic connection is potentiated if a presynaptic spike fires ahead of a postsynaptic one and is depressed otherwise [19]. There is also “reverse” or “anti-”STDP, which produces the opposite behavior [20,21,22]. Known multi-component models of in vitro neural cultures can form different burst patterns that mimic different stages of the culture’s development and generate network bursts similar to those observed experimentally (see, e.g., [14,23,24,25]).

The investigation of cognitive maps, i.e., mental representation of the environment, has been a hot topic in Neuroscience for more than 70 years [26,27]. The neurobiological bases have been put forward in pioneering works by O’Keefe and Mosers. For example, they discovered that the animal’s position in the environment determines the activity of the so-called place and grid cells [28,29]. Several mathematical models simulating the emergence of cognitive maps by associating external stimuli with internal (neuronal) representations have been proposed (see, e.g., [30,31,32,33]). However, the implication of Hebbian plasticity in this process is an open issue.

Ponulak and Hopfield proposed a model with an anti-STDP rule that leads to the ability of an SNN to “remember” the localization of a “positive” stimulus [34]. In such an SNN, synaptic connections directed to the stimulation locus are potentiated. As a result, the activation of distant neurons causes a wave of spikes traveling to the neurons responsible for encoding positive stimuli. We have recently shown that stimulation of an SNN with STDP potentiates synaptic couplings, which directed the stimulus locus outwards [35,36,37]. Thus, for the plane STDP, the vector field, created by synaptic connections, diverges from the stimulation point. We can then hypothesize that STDP is responsible for codifying “negative” or dangerous stimuli and enables the representation of obstacles in cognitive maps.

In this work, we propose a solution to spatial memory representation in an STDP-driven SNN. In the model, the information is codified by the location of the stimulation locus. Memory retrieval occurs through the readout of the functional response in the form of population bursts. Note that the memory acquisition is not entirely reliable due to ongoing spontaneous activity that can “erase” spatial traces of previous stimuli. Using the synaptic vector field concept, we propose a measure of the global network memory and analyze its relation to learning curves. We show that the time of evoked network synchronization depends on the stimulus’s number. To validate the approach, we simulate a wheeled robot driven by the SNN and show that it successfully maps regions in the environment associated with dangerous experiences. Neurons in the SNN play the role of place cells, and the waves of spikes traveling in the network determine the direction of the robot’s movements. As a result, the SNN implements a functional cognitive map with a robot embodiment, which helps the robot avoiding dangerous zones.

## 2. Materials and Methods

We used the phenomenological Izhikevich model to simulate the dynamics of the neuronal membrane [38]. In terms of biological relevance, it is similar to the Hodgkin–Huxley model; however, it requires significantly fewer computational resources, especially when modeling large neural networks [39]. The following dynamical system describes the model:(1)$$\frac{dv}{dt}=0.04v^{2}+5v+140-u+I(t),$$
(2)$$\frac{du}{dt}=a(bv-u),$$
with the additional condition to reset the variables when the spike peak is reached:(3)$$\mathrm{if}\;v\geq +30\;\mathrm{mV},\;\mathrm{then}\;\left\{ \begin{array}{l} v\leftarrow c\\ u\leftarrow u+d \end{array}\right.,$$
where *v* is the transmembrane potential, *u* is the recovery variable, *a*, *b*, *c*, *d* are the parameters, and *I*(*t*) is the external current. When the potential reaches 30 mV, a spike is recorded, and the variables are reset to the values described in the equations. We set: *a* = 0.02; *b* = 0.2; *c* = −65 and *d* = 8, which, in the absence of external input, cause the neuron to remain at rest, while an external direct current produces regular spikes typical for cortical neurons [38,39].

In Equation (1), the external current is given by:(4)$$I(t)=\xi (t)+I_{syn}(t)+I_{stml}(t),$$
where *ξ*(*t*) is an uncorrelated Gaussian white noise with zero mean and standard deviation *D*, *I_syn_*(*t*) is the synaptic current, and *I_stml_*(*t*) is the external stimulation current. The external stimulus was delivered as a sequence of pulses with a frequency of 10 Hz, duration 3 ms, and amplitude sufficient to excite the neuron. We applied stimulation until the onset of the network synchronization effect (see below); its duration was from 20 s (with instant synchronization, possible after the repeated stimulation) to several minutes.

The synaptic current was calculated as the weighted sum of the output signals of neurons sending connections to a given neuron:(5)$${I_{{}^{i}}}_{syn}(t)=\sum_{j}g_{j}w_{ij}y_{ij}(t),$$
where *g_j_ is* the transformation coefficient of the output signal of neuron *j* into synaptic current (we set *g* = 20 for excitatory and *g* = −20 for inhibitory neurons), *w_ij_* is the weight of the coupling going from the presynaptic neuron *j* to the postsynaptic neuron *i*, and *y_ij_*(*t*) is the output signal of a presynaptic neuron, which is understood as a portion of a neurotransmitter emitted at synapses with each pulse. The dynamics of the mediator was determined by the Tsodyks–Markram model, which takes into account the effects of short-term synaptic plasticity [40]:(6)$$\frac{dx_{ij}}{dt}=\frac{z_{ij}}{\tau_{rec}}-u_{ij}^{*}x_{ij}\delta (t-t_{j}-\tau_{ij}),$$
(7)$$\frac{dy_{ij}}{dt}=-\frac{y_{ij}}{\tau_{I}}+u_{ij}^{*}x_{ij}\delta (t-t_{j}-\tau_{ij}),$$
(8)$$\frac{dz_{ij}}{dt}=\frac{y_{ij}}{\tau_{I}}-\frac{z_{ij}}{\tau_{rec}},$$
(9)$$\frac{du_{ij}^{*}}{dt}=\frac{u_{ij}^{*}}{\tau_{facil}}+0.5(1-u_{ij}^{*})\delta (t-t_{j}-\tau_{ij}),$$
where *x_ij_, y_ij_, z_ij_* are the parts of the synapse mediator connecting neuron *j* with neuron *i*, which are in a restored, active, and inactivated state; *t_j_* is the pulse generation time by the presynaptic neuron, determined by Equation (3); $\tau_{I}$, $\tau_{rec}$ and $\tau_{facil}$ are the characteristic times of the processes of inactivation, restoration, and facilitation; $\tau_{ij}$ is the axonal delay in the arrival of a spike to the synaptic terminal; $u_{ij}^{*}$ is the part of the mediator released from the available reserve *x_ij_* at each spike. The parameter values used in this study allowed synapses to exhibit both the effects of depression (in the case of high-frequency activity) and facilitation (in the case of activity with a frequency of about 1 Hz): $\tau_{I}$ = 10 ms, $\tau_{rec}$ = 50 ms and $\tau_{facil}$ = 1000 ms. Axonal delays were proportional to the distances between neurons (see below).

Long-term synaptic plasticity, as a basis of the global network memory, was represented by STDP. To simulate STDP, we used an algorithm with local variables [19]:(10)$$\frac{ds_{i}}{dt}=-\frac{s_{i}}{\tau }+\delta (t-t_{i})$$
(11)$$\frac{ds_{j}}{dt}=-\frac{s_{j}}{\tau }+\delta (t-t_{j}-\tau_{ij})$$
(12)$$\frac{dw_{ij}}{dt}=F_{+}(w_{ij})s_{j}(t)\delta (t-t_{i}^{})-F_{-}(w_{ij})s_{i}(t)\delta (t-t_{j}^{}-\tau_{ij}),$$
where *s_i_* and *s_j_* are variables that track spikes on the postsynaptic and presynaptic neurons, respectively, *τ* = 10 ms is the characteristic decay time of local variables, *t_i_* and *t_j_* are the time of spike generation on the postsynaptic (receiving spikes) and presynaptic (transmitting spikes) neuron. In turn, the functions of weight increase and decay followed the multiplicative rule [19,41]:*F*_+_(*w_ij_*) = *λ*(1 − *w_ij_*),(13)
*F_−_*(*w_ij_*) = *λαw_ij_*,(14)
where *λ* = 0.001 is the learning rate, and *α* = 5 is the asymmetry parameter that determines the ratio of the processes of weakening and strengthening of the synapse.

To model the spatially extended SNN, we randomly distributed on a 1.2 × 1.2 mm square 400 excitatory and 100 inhibitory neurons. The probability of connection decreased with the distance between neurons. The speed of the spike propagation along an axon was set to 0.05 m/s. Thus, the maximal axonal latency, $\tau_{ij}$, between two neurons located in different corners of the network was 34 ms. However, when the excitation propagates in the form of a traveling wave, its velocity is also determined by synaptic latencies, that is, the time required for the transformation of the excitatory postsynaptic current into the postsynaptic potential and the generation of a spike.

The spatial directionality of connections between neurons was visualized by a vector field (Figure 1), which was constructed as follows: (1) The area occupied by the modeled neural network was divided into even cells. Depending on the tasks (see below), a different lattice spacing was used. (2) Each connection between neurons was represented as a vector, the direction of which coincided with the vector connecting the neurons, and the length was equal to the synaptic weight of the connection *w_ij_*. (3) For each cell, we calculated the resultant vector of connections. This calculation incorporated all links geometrically passing through the current cell. The resulting cell vector was calculated using the vector sum of the link vectors. (4) All the resulting cell vectors were represented by arrows. In this case, the beginning of the arrow was set in the center of the cell, the direction coincided with the direction of the resultant vector, and the length was proportional (with the visualization coefficient) to the length of the vector.

In addition to the vector field describing synaptic (anatomical) connections, we also used the vector field of neural activity, i.e., functional connections or connectomes. The direction of the vector of the neural activity coincides with the synaptic vector. However, its length was determined by the history of spikes passed through this connection and that excited postsynaptic neurons:(15)$$\frac{dl_{ij}}{dt}\;=cy_{ij}\delta \left(t-t_{sp}^{i}\right)-\;\frac{l_{ij}}{\tau_{l}},$$
where *l_ij_* is the length of the connection activity vector going from neuron *j* to neuron *i*, *c* is the activation constant, *y_ij_* is the output of presynaptic neuron *j*, *t^i^_sp_* is the time instant of postsynaptic spike *i*, and $\tau_{l}$ is the relaxation time.

We performed the numerical integration of the model (1)–(15) using the Euler method with a step of 0.5 ms. Such a procedure has been shown to be appropriate for integrating large systems of the Izhikevich’s neurons [38,39]. We developed a custom software platform in C++ using the cross-platform IDE QT, called NeuroNet, which enables online simulation of the model, building the vector fields (Figure 1), and implementing the robot environment and control (see below).

## 3. Results

### 3.1. Synaptic Memory at Network Scale: Pathways of Spike Patches Match Potentiated Neural Couplings

The characteristic feature of both the model network and living cultures of neurons grown in vitro is the burst-like nature of their activity. In addition, each population burst of spikes topographically corresponds to a wave or a patch of activity traveling in a network monolayer. The results of experiments and simulations allow us to hypothesize that such a moving patch of excitation plays the role of the central functional unit in the brain’s information processes [42,43]. Model simulations have shown the possibility of the formation of associative connections caused by the interaction of traveling waves [44]. Earlier, we have shown that STDP facilitates the propagation of traveling waves [35,36]. On the network scale, it can be used in associative learning based on the potentiation of the shortest neural pathways and depression of alternative longer routes [45].

Figure 2 illustrates the emergence of a vector field of functional connections during the propagation of a traveling patch in the form of a population burst. As expected, the vector field coincides with the primary direction of the patch propagation. After a long-term external stimulation (Figure 2B), the spontaneous activity mostly repeats the activity triggered by the stimulation (Figure 2C). Thus, the SNN remembers the external stimuli and reproduces similar activity patches even after the stimulation has been stopped.

Let us now study the origin of the moving-patches memory. Figure 3 shows the vector fields of functional (red arrows) and anatomical (black arrows) connections and their differences (intensity of the magenta color). Both fields generally coincide, although some deviations can be observed. The differences appeared mainly in areas where a significant weight rearrangement occurred during external stimulation (Figure 3B). Thus, the network functionality in the form of spike waves reveals the structure of interneuron connections. The vector fields before (Figure 3A) and after (Figure 3C) external stimulation showed STDP-dependent changes in the inter-neuronal connections. In particular, the connections originating in the stimulation zone were strengthened, facilitating the propagation of spike patches traveling from this location. Further on, to study the properties of global spatial memory, we mainly considered the vector field of synaptic connections, while in the experiments with a robot, we paid attention to the field of functional couplings.

### 3.2. Stimulation of SNN Is an Iteration of Recording into Network Memory

Let us take a closer look at how the STDP-mediated network restructuration occurs. At the beginning of the stimulation of a region of the SNN (Figure 4A), the evoked activity was highly variable. Stimulus pulses did not always excite population bursts. Furthermore, their duration and amplitude fluctuated significantly (Figure 4A, bottom panel). The prolonged exposure to stimulation prompted an STDP-mediated enhancement of the connections originating in the stimulated region. On the single neuron scale (zoomed regions in Figure 4), potentiation of synapses occurred for couplings coming from the locations on the stimulus side. Such an increase is explained by the ability of STDP to strengthen the connections that conducts first spikes within a compound spikes train [46,47]. The vector field of synaptic connections can be used to visualize memory traces. By scaling up the vector lattice, we can estimate the global connectivity vector ***g****_c_*, representing the average (over the first quadrant) coupling direction and the global memory state (towards the stimulus location in Figure 4A).

The change in the direction of the anatomical vectors indicated a rearrangement of the synaptic weights (Figure 4B, top panel). In turn, strengthening centrifugal connections facilitated the propagation of spiking patches from the stimulus location. This led to the synchronization between the stimulus and network response. Each applied stimulus caused the same population burst. The burst frequency became equal to the stimulation rate (Figure 4B, bottom panel). We used such a frequency lock to automatically detect the beginning of synchronization while determining the time required for synchronization [36,48]. At the same time, the vector of global network memory changed its direction to the stimulus location outwards (Figure 4B, thick red vector ***g****_st_*), and we can speak about storing stimulus information on the global network scale.

When the stimulation stops, the synaptic weights remained unchanged for a certain time. The repeated activation of the stimulation triggered synchronous responses. When another zone of the network was stimulated, such fast synchronization was no longer observed (Figure 4C). Thus, the STDP-mediated structural network rearrangement can memorize the localization of the applied stimulation.

Over time, spontaneous activity led to rearrangements of the connectome. Accordingly, the resumed stimulation initially did not trigger synchronous population bursts. Thus, besides storing information about external stimulation, the SNN can also “forget” it over time. Then, the global connectivity vector ***g*** can be used to quantify memory states. In particular, we introduced the following measure of global network memory as the cosine of the angle between the current connectivity vector ***g****_c_*(*t*) and the vector after stimulation ***g****_st_*:(16)$$M=\frac{\left(g_{c},g_{st}\right)}{\left\|g_{c}\right\|\left\|g_{st}\right\|}.$$

The value *M* = 1 corresponds to perfect memorization of the stimulus location (it is observed, e.g., immediately after the stimulation), while *M* = −1 describes the case of complete forgetting.

Thus, the formation of global memory in the SNN occurred by rearrangement of synaptic couplings under stimulating a region of the network. It allowed recording the stimulus location. Memory retrieval was performed by detecting a functional, synchronized response of the network to a stimulus.

### 3.3. Learning and Forgetting Spatial Stimuli

The structural and functional interactions that underlie memory formation and retrieval can be studied by external stimulation repeated after long enough time intervals. Figure 5A shows the learning curves obtained by stimulating the SNN ten times with inter-stimulation intervals of about $10^{4}$ s. Repetitive stimulation provoked the acceleration of functional response in the form of synchronization. The relative response time followed a power-law dependence (blue curve). The temporal variability required to achieve a response was especially pronounced during the second and third stimulations. We also observed that the network memory measure *M* followed a power law (magenta curve), which allowed us to assume that the functional response time correlates with the residual global network memory.

We note that learning curves (Figure 5A) were sensitive to the intervals between stimulations. The acceleration of the functional response with the stimulus number disappeared if the interval between stimulations or neural noise level increased (Figure 5B, blue). At the same time, there was (almost) no change in the residual global memory *M*, which stayed in the negative zone (Figure 5B, magenta). This result indicated that the structural rearrangements elicited by previous stimuli were forgotten and hence, did not accumulate.

Figure 5C displays all pairs (Synchronization time, *M*) found for different parameter sets. It revealed a correlation between these characteristics. For the same values of the residual network memory *M*, synchronous response time could vary strongly. The smaller *M*, the larger fluctuations of the synchronization time. The value of *M* determined the maximal possible time to reach a functional response. Such dependence can be explained by the presence of neural noise and, accordingly, by the stochastic nature of STDP rearrangements [36]. We also calculated *M* for one quadrant of the network only and did not consider the weights rearrangements in the other three quadrants. However, we noted that higher residual global memory values always corresponded to shorter functional response times.

Thus, the network memory ultimately determines the decay of the learning curves. In other words, a falling learning curve indicates that even when the next stimulation begins, there remains a certain proportion of synaptic couplings shaped by the previous stimulation. Since training a neural network is iterative, it is evident that the forgetting rate—the disappearance of network memory traces—is an extremely significant characteristic. A decaying learning curve shows that memory traces’ storage duration is higher than the time between stimulus iterations.

### 3.4. Spatial Memory with Negative Reinforcement: Embodiment of the SNN in a Robot

The brain continuously learns and works under active interaction with the environment. Neuro-robotics provide the means for testing artificial neural networks in a similar context. Earlier, several approaches to controlling mobile robots have been proposed (see, e.g., [45,49,50,51,52,53]). Let us now adopt such a strategy and study how a robot can use global spatial memory to learn dangerous zones in the environment.

We embodied the SNN into a robot moving in a square arena (Figure 6A). In this context, neurons play the role of place cells whose receptive fields are stimulated when the robot passes through the corresponding areas in the arena (Figure 6B). Receptive fields of nearby neurons overlap. Each robot’s location in the environment corresponds to a circle with the radius *r* = 40 µm in the network space that receives stimulation. While the robot remains in a safe zone, the stimulation of the receptive fields occurs at 1 Hz. If the robot gets into an area with negative reinforcement, the stimulation frequency increases to 10 Hz. Thus, the SNN interprets the external environment from the allocentric viewpoint.

We expect that place cells will learn to distinguish between the safe and dangerous zones in the arena. The spiking activity of the SNN determines the robot behavior (Figure 6C). The activity vector (functional connectome) evaluated at the current robot position defines the robot movement’s direction and speed. Thus, in the arena, the robot tries to follow spike patches traveling in the network space.

In simulations, the dangerous zone occupied 25% of the total area in different locations (e.g., the III quadrant in Figure 6A). First, we performed experiments without learning, i.e., STDP was switched off. The robot spent on average 42.2% ± 3.2% of the total time in the danger zone (the number of experiments *n* = 18). The discrepancy between the area and the time spent in the dangerous region was due to the robot’s velocity. When the robot was in the danger zone, the stimulation frequency was high (10 Hz). Then, the functional connections rapidly changed, and as a result, the length of the functional vector defining the robot velocity was relatively small. Thus, the mean robot velocity in the dangerous area was significantly lower than in the safe part of the arena. Therefore, the robot was moving in the dangerous part about twice as long as expected. Second, we tested the learning abilities of the robot by switching on the STDP. In this condition, the time spent by the robot in the danger zone decreased to 21.9% ± 2.1%, which tightly correlated with its area. Third, we turned off the STDP again to measure the learning performance. Then, the average time spent by the robot in the danger zone decreased to 7.9% ± 0.7%. Such a significant decrease (42.2% vs. 7.9%) confirms the ability of the SNN embodied in a robot to create a cognitive map of the environment.

Let us now study the robot’s ability for relearning, i.e., the capacity to adapt its behavior to changing environments. Figure 7A shows the robot movement in the arena without dangerous zones. As expected, the robot exhibited no particular preferences for the arena segments. In particular, it spent about 25% of the time in quadrants I and III (blue and red curves in the bottom subplot of Figure 7A). Then, we turned on the danger zone in quadrant III. After learning, the robot preferred staying away from the dangerous area (Figure 7B). The amount of time spent in quadrant III significantly decayed with time (red curve in Figure 7B). We also noted that such an avoidance behavior was not complete, i.e., the learning was not “perfect.” From time to time, the robot “forgot” about its “negative experience” and entered into the danger zone, which is compatible with animals’ typical explorative behavior.

Such an “improper” behavior of the robot induced by the SNN-based cognitive map allows relearning in the case of environmental changes. To illustrate this, we moved the danger zone to quadrant I (i.e., quadrant III became safe). Figure 7C shows that after a while, the robot started avoiding the dangerous quadrant I and learned that the III one was now safe and could be explored. Time spent in quadrant III raised on average to 34%. Relearning was possible since the robot eventually entered into quadrant III and observed that it was not dangerous anymore. We also noted that relearning took about twice as long time as initial learning. The longer time is explained by the necessity of rearranging synaptic weights in the SNN, which agrees with our previous results for small neural ensembles [45].

## 4. Discussion

In this work, we studied the emergence of global spatial memory in a spiking neural network. We showed that the initially unstructured spatially extended SNN equipped with STDP performed specific structural and functional rearrangements and eventually memorized spatially distributed stimuli. Such a form of representation of information is a particular case of the more general coding by channel number, widely extended in animals’ central nervous system [54]. The discussed spatial coding may underlie cognitive maps responsible for the spatial navigation of animals and humans [34,55].

To describe the spatial properties of the SNN, we used functional and anatomical vector fields. In terms of the anatomical vector field, the stimulation locus corresponded to a source of the field of synaptic connections. To quantify the network memory state, we introduced a novel measure. It describes the difference between the current memory vector and the vector obtained after stimulation. The memory vector represents the mean direction of synaptic connections in a big area of the network. The measure allows quantifying the level of remembering by the network of the stimulus applied before.

The recording of spatial information into memory occurs through a local stimulation of a part of the SNN. The stimulation changes the network connectome. Memory retrieval can be performed by detecting a functional response of the SNN synchronized with an external stimulus. We can also monitor the synaptic support of memory, which helps study network memory’s fundamental laws. Over time, spontaneous network activity leads to the rearrangement of anatomical connections, and the connectome returns to its initial state. Thus, the SNN can store information about external stimuli and also forget it over time.

Repetitive stimulation of the SNN speeds up the functional response to each subsequent stimulus. The obtained learning curve and the global memory measure followed power laws. We showed that the global memory, remaining after previous stimulations, determines the maximal delay of a functional response. Thus, a correct (falling) learning curve can be observed if the forgetting process does not empty the memory. In experimental studies with neural cultures grown in vitro, repetitive stimulation also led to falling learning curves. However, following our results, this effect of accelerating the formation of a functional response was not always observed [9,10]. Thus, we hypothesize that the reported in the literature differences in the individual “patterns” generated by neuronal cultures can be explained by a hidden parameter related to the network connectivity level.

Learning curves obtained in various psychological studies in humans and behavioral experiments with animals, as a rule, follow “the power law of practice” [4,5]. To explain such a dependence, the theory of hierarchical memory fragments has been put forward [56]. Each memory fragment helps solve the problem at its level, and, at the same time, it may include a link to another fragment at a lower level. Our data indicated that a system using global network memory based on local Hebbian plasticity requires no hierarchical fragmentation. It can manifest an adaptive response and power-law learning curves for free. Experimental learning curves can originate from the accumulation of residual network memory, provided that the interval between stimuli is below the forgetting time. In this case, the power-law dependence can result from a simple summation of various individual curves, which was demonstrated earlier for model memory systems of a formal nature [57,58].

Finally, we validated the functionality of spatial memory by embodying the SNN into a roving robot. The robot moved in an arena divided into safe and dangerous zones. The SNN controlled the robot’s direction and velocity. We showed that the SNN could effectively build a cognitive map of the arena, and consequently, the robot started avoiding the dangerous region. Thus, the robot exhibited a basic spatial cognition linked to “his/her” adaptive behavior. However, the learning was not perfect. After training, the robot eventually entered into the dangerous area, as animals do in nature. Such a behavior can be seen as a drawback from the viewpoint of standard AI neural networks. Though, it turns into an advantage. If the environment changes and the dangerous zone moves into another place, the SNN can remap positive and negative zones. It happened since the robot eventually visited the previously dangerous area and observed that now it was safe and could be explored, which provided the experience required for remapping. Thus, in contrast to the “catastrophic interference” phenomenon known for some AI architectures, imperfect learning is the basis for adapting the subject to the time-evolving world.

## Figures and Tables

**Figure 1 sensors-21-02678-f001:**
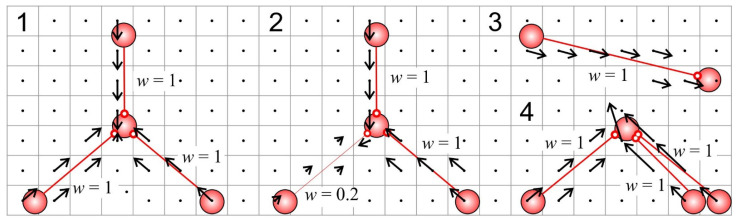
Examples of vector fields defined by synaptic connections in four small neural assemblies. Black arrows show the direction and magnitude of the synaptic vector field in a grid containing 7 × 23 cells (black points correspond to zero vectors). The thickness of the red links connecting neurons is proportional to the weight of interneuron couplings.

**Figure 2 sensors-21-02678-f002:**
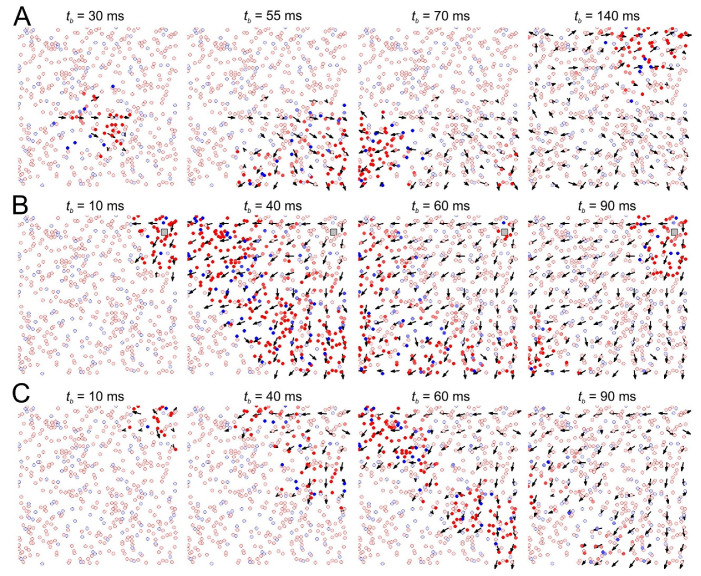
Emergence of traveling patches of neural activity and their relation with the vector field of functional connections (black arrow). (**A**) Spontaneous firing before stimulation (red and blue circles mark excited excitatory and inhibitory neurons, respectively; the time instants *t_b_* of the network burst refers to the interval past from the stimulus onset). (**B**) Network activity during periodic local stimulation. (**C**) Spontaneous activity after long-term stimulation.

**Figure 3 sensors-21-02678-f003:**
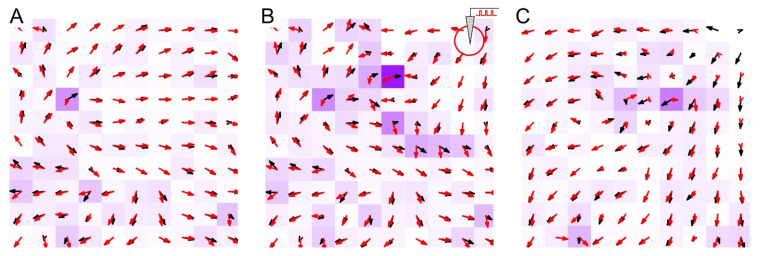
Comparison of the vector fields of anatomical (black arrows) and functional (red arrow) connections during neuronal activity before (**A**), during (**B**), and after (**C**) a period of long-term local stimulation. The intensity of the magenta color of the cells is proportional to the field difference.

**Figure 4 sensors-21-02678-f004:**
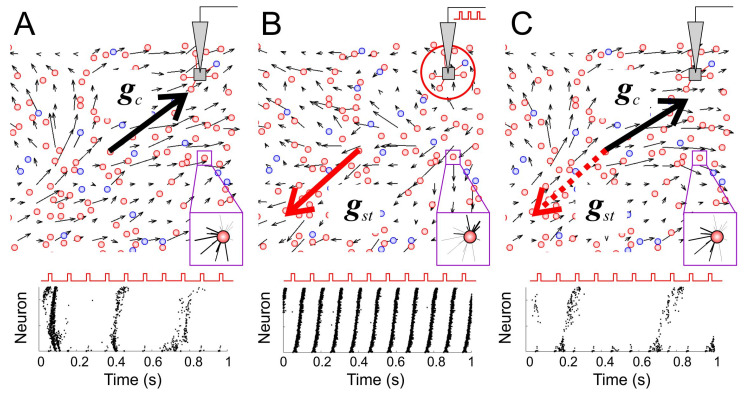
Characteristics of global network memory. (**A**) *Top*: The vector field of the synaptic weights (thin black arrows) of a segment of the neural network before external stimulation (only the first quadrant of the network is shown). The thick black arrow, ***g****_c_*, shows the global connectivity vector of the selected segment. The magenta-framed inset shows incoming couplings for a representative neuron (line thickness corresponds to the coupling weight). *Bottom*: A typical epoch of the raster plot during the stimulation (each point displays a spike of the corresponding neuron; neurons have been sorted by their distance to the stimulation site). (**B**) Same as in (**A**) but immediately after external stimulation. The thick red arrow, ***g****_st_*, is the vector of global memory after stimulation. (**C**) Same as in (**A**) but after a period of forgetting. The cosine of the angle between the connectivity vector at the current time ***g****_c_* (black) and the vector after stimulation ***g****_st_* (dashed, red) is the measure of global network memory.

**Figure 5 sensors-21-02678-f005:**
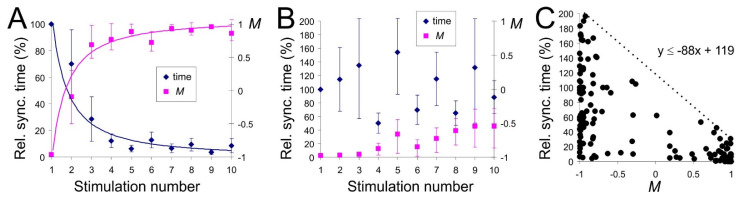
Correlation of the learning curve with the global network memory measure. (**A**) Averaged relative synchronization time (learning curve) and memory measure *M* as functions of the stimulus number. The time interval between successive stimulations is $10^{4}-3\times 10^{4}$ s, the noise intensity *D* = 4.8 (*n* = 10). The relative synchronization time is the time required to synchronize the network, normalized to the synchronization time at the first stimulation. *M* was evaluated just before the stimulation. (**B**) Same as in (**A**) but for the increased time interval between stimulations ($5\times 10^{4}-10^{5}$ s) and the intensity of neural noise (*D* = 5.1–5.5; *n* = 10). (**C**) Synchronization time versus the global memory measure (*n* = 200).

**Figure 6 sensors-21-02678-f006:**
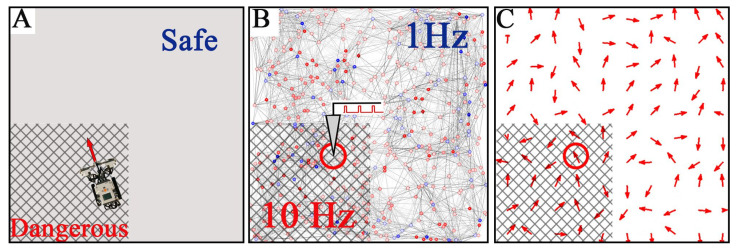
Embodiment of the SNN in a moving robot. (**A**) The robot is controlled by the SNN and moves in a square arena divided into safe and dangerous (marked with a grid) zones. (**B**) The robot location determines the stimulation area in the network space (red circle). The stimulation frequency is 1 and 10 Hz for the safe and dangerous zones, respectively. (**C**) The vector field of functional connections (red arrows) in the SNN determines the robot’s direction and speed (red circle delimits the area controlling the robot movement).

**Figure 7 sensors-21-02678-f007:**
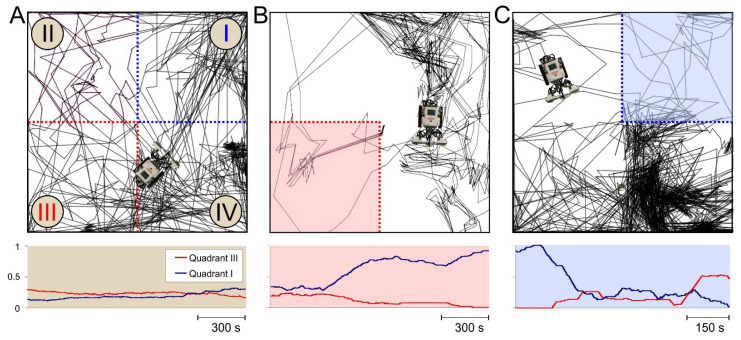
Adapting the robot behavior to changing environment. (**A**) Top: Traces of the robot movements in the arena without dangerous zones (all four quadrants are safe). The robot randomly explored the environment. Bottom: Moving average of the time spent by the robot in quadrants I (blue curve) and III (red curve). (**B**) Same as in (**A**) but during movement in the arena with the danger zone in quadrant III. The robot spent little time in the dangerous quadrant (red curve), while the time spent in quadrant I rose (blue curve). (**C**) Same as in (**B**) but during relearning. The dangerous zone had moved from quadrant III to I. The robot learned the new dangerous area and recovered the previous one that was now safe. The times spent by the robot in quadrants I and III were inverted (the blue curve decayed, while the red one increased).

## Data Availability

Not applicable.

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
