# Peer review of "Spatial Memory in a Spiking Neural Network with Robot Embodiment"

_sensors, 2021, doi:10.3390/s21082678_

Round 1
Reviewer 1 Report
This paper presents a spatially randomly distributed spiking neural network to define regions within the map. The learning process used is STDP and the main contribution is the utilization of the visualization of the synaptic field vector as a memory representation.
In the abstract the problem to be solved is defined as the solution of a maze. This may cause confusion to the reader since a trajectory is not solved to solve a maze (as in other similar works) but rather the network is trained to detect safe zones.
As stated in the introduction, memory shows a time dependence that leads to forgetting of learning. For this reason I miss the concept of "Catastrophic interference" and the solutions that have been given in the past.
In the explanation of STDP in eq. 12 LTP and LTD parameters are not explained.
Figure 1 should be explained in more detail, it is the first time that the visualization with vectors is presented and it can generate confusion. It should be commented that 14 neurons (circles) are being viasualized and that the space is discretized in a 7x23 grid, calculating the vector in each center.
In spatial learning, despite having a considerable number of neurons, not all the vectors point out of the dangerous area. This means that the network is not fully trained as in my opinion the detection problem is not completely solved. This is visible in Figure 7C where trajectories within the dangerous area are evident as already predicted by the vectors. The network should be trained in depth and tested with at least one other dangerous zone within the map to demonstrate the capabilities of the method.
Author Response
We thank the reviewer for careful reading of the manuscript and deep questions that helped us improve the manuscript. Below, please find a point-by-point account for the raised critics.
In the abstract the problem to be solved is defined as the solution of a maze.
Corrected.
As stated in the introduction, memory shows a time dependence that leads to forgetting of learning. For this reason I miss the concept of "Catastrophic interference" and the solutions that have been given in the past.
We thank the referee for pointing us to this important issue. In the new version of the ms, we have addressed it. Briefly, the imperfect learning allows for relearning of spatial maps and adapting the robot behavior to changing environments (see Figure 7 for more details).
In the explanation of STDP in eq. 12 LTP and LTD parameters are not explained.
All parameters and their values are given in the main text (see after equation (12) and (13-14)).
Figure 1 should be explained in more detail, it is the first time that the visualization with vectors is presented and it can generate confusion. It should be commented that 14 neurons (circles) are being viasualized and that the space is discretized in a 7x23 grid, calculating the vector in each center.
Thank you. We have added this information.
In spatial learning, despite having a considerable number of neurons, not all the vectors point out of the dangerous area. This means that the network is not fully trained as in my opinion the detection problem is not completely solved. This is visible in Figure 7C where trajectories within the dangerous area are evident as already predicted by the vectors. The network should be trained in depth and tested with at least one other dangerous zone within the map to demonstrate the capabilities of the method.
We agree with the reviewer. The network never learns the location of the dangerous zone with a 100% fidelity. Note, however, that this provides an advantage to the robot (similar to the behavior of animals). It allows effective relearning in changing environments. We have made additional simulations and extended our results shown in Figure 7.
Reviewer 2 Report
It was my pleasure to read and review this well-written and inspiring manuscript. The considered topic is extremely relevant in nonlinear neuroscience. I especially value the application of the developed approach to real robot control. However, I have several questions and recommendations I want to address to the authors. Please, find them below:
- Please, clarify the choice of the Izhikevich neuron model. Besides there are a lot of simplified Hodgkin-Huxley models known, Izhikevich model has plenty of shortcomings in comparison with the original HH model.
- Considering the computational costs, the choice of numerical integration method is of great importance. Which numerical methods did you use in your experiments? Please, indicate the used software as well.
- It is known that numerical methods can seriously affect the behavior of nonlinear systems, especially in large-scale networks. How this fact is taken into account in your study?
- Did you compare the proposed approach to robot control with any conventional control methods? I believe the described task can be performed without complex neuromorphic systems. How conventional ANNs solve this task? Any comparison? I also recommend choosing a more complex and novel (see question 5) physical system to control in your further studies.
- Please, clarify the contribution of the current paper after your previous papers:
- Lobov, S.A.; Mikhaylov, A.N.; Shamshin, M.; Makarov, V.A.; Kazantsev, V.B. Spatial properties of STDP in a self-learning spiking neural network enable controlling a mobile robot. Front. Neurosci. 2020, 14, 88.
- Zharinov, A. I., Makarov, V. A., Kazantsev, V. B., & Lobov, S. A. (2020). Spatial memory based on an STDP-driven neural network. 2020 4th Scientific School on Dynamics of Complex Networks and Their Application in Intellectual Robotics (DCNAIR).
As far as this Reviewer sees, the Introduction of the paper is the deeply reworked text from paper (b). Moreover, the experimental part of the reviewed manuscript declares "The robot spent in the danger zone on average 48.1% ± 2.8% of the total time (n = 11)" which repeats the text from paper (b), where the experimental set supposed to be different from present paper. Thus, did you obtain precisely the same results in two different experimental runs, or this experiment is completely reproduced from (b) too? Anyway, any fact of self-citation should be clearly indicated in the revised manuscript with proper referencing.
Some minor remarks:
"waves of spikes travelling " - possible typo ("traveling")
Nevertheless, I like this comprehensive study and can recommend it for publication after relatively moderate revisions.
Author Response
We thank the reviewer for careful reading of the manuscript and deep questions that helped us improve the manuscript. Below, please find a point-by-point account for the raised critics.
- Please, clarify the choice of the Izhikevich neuron model. Besides there are a lot of simplified Hodgkin-Huxley models known, Izhikevich model has plenty of shortcomings in comparison with the original HH model.
In general, we believe that our results do not depend on the choice of the model. On the one hand, the used model reproduces the key neurophysiological properties of biological neurons (DOI: 10.1109/TNN.2004.832719). On the other, it is computationally efficient compared to the classical Hodgkin-Huxley model. The latter is especially important when dealing with large neural networks.
2. Considering the computational costs, the choice of numerical integration method is of great importance. Which numerical methods did you use in your experiments? Please, indicate the used software as well.
We provided this information in the new version of the manuscript.
3. It is known that numerical methods can seriously affect the behavior of nonlinear systems, especially in large-scale networks. How this fact is taken into account in your study?
The software has been thoroughly checked for reproducibility of the results in our previous works.
4. Did you compare the proposed approach to robot control with any conventional control methods? I believe the described task can be performed without complex neuromorphic systems. How conventional ANNs solve this task? Any comparison? I also recommend choosing a more complex and novel (see question 5) physical system to control in your further studies.
We thank the reviewer for this deep question, which can be addressed in an additional research. Here we add that our present work aims at two complementary goals. On the one hand, we tried to provide a biologically inspired model capable of controlling the robot behavior in a way as animals do. On the other hand, we aimed at showing that learning and forgetting are two inseparable principles existing in animals. As we show in the manuscript, learning should not be perfect. The chance to make a “mistake” provides a crucial difference between the standard ANN and our model. So the direct comparison is not fair. To stress this issue, we performed additional experiments and significantly extended Figure 7.
5. Please, clarify the contribution of the current paper after your previous papers:
a. Lobov, S.A.; Mikhaylov, A.N.; Shamshin, M.; Makarov, V.A.; Kazantsev, V.B. Spatial properties of STDP in a self-learning spiking neural network enable controlling a mobile robot. Front. Neurosci. 2020, 14, 88.
b. Zharinov, A. I., Makarov, V. A., Kazantsev, V. B., & Lobov, S. A. (2020). Spatial memory based on an STDP-driven neural network. 2020 4th Scientific School on Dynamics of Complex Networks and Their Application in Intellectual Robotics (DCNAIR).
As the reviewer noticed, the current paper stands on the shoulders of our previous works. With paper (a) the current research shares the use of the model of synaptic plasticity and robot for the testing purpose (though with completely different goals). So, no significant overlap exists.
Paper (b) is a conference work, which established preliminary results. In the current manuscript, first, we increased statistics. Second, we reoriented the manuscript to the study of network memory. Third, we provided a novel viewpoint onto imperfect learning. Fourth, we showed the capacity of relearning in cognitive maps and tested it in robots. So, again there is no much overlap.
"waves of spikes travelling " - possible typo ("traveling")
Fixed.
Round 2
Reviewer 2 Report
Thank you for providing a revised version of your manuscript. All my questions were properly answered and the paper was significantly improved during the revision. Minor polishing of the text is still needed, nevertheless, I can recommend the revised manuscript for publication in MDPI Sensors and wish the authors all the best in their future studies.
Author Response
We thank the reviewer again for careful reading of the manuscript and deep questions that helped us improve the manuscript.